# Waterborne Gastrointestinal Diseases and Child Mortality: A Study of Socioeconomic Inequality in Mexico

**DOI:** 10.3390/ijerph21111399

**Published:** 2024-10-23

**Authors:** Jorge Armando Morales-Novelo, Lilia Rodríguez-Tapia, Carolina Massiel Medina-Rivas, Daniel Alfredo Revollo-Fernández

**Affiliations:** Department of Economics, Metropolitan Autonomous University, Campus Azcapotzalco, Av. San Pablo 420, Col. Nueva El Rosario, Alcaldía Azcapotzalco, México City 02128, Mexico; jamn@azc.uam.mx (J.A.M.-N.); cmmr@azc.uam.mx (C.M.M.-R.); darf@azc.uam.mx (D.A.R.-F.)

**Keywords:** waterborne gastrointestinal diseases, child mortality, water access and mortality, public policies and child mortality, health inequities, mortality modeling

## Abstract

In Mexico, 1.9% of child mortality among children aged 3 to 15 years is attributed to waterborne gastrointestinal diseases (WGD). This study employs a generalized bivariate logit econometric model to simulate the relationships between mortality risks and seven explanatory variables. Based on the model results and sensitivity analysis of the estimated parameters, a set of policies was designed to reduce the likelihood of child mortality. The proposed strategy involves implementing the following public policies, primarily targeting communities with extreme and high marginalization: increasing access to drinking water, improving housing conditions, expanding parental basic education coverage, and providing nutrition and healthcare to children from an early age. The findings reveal that children who speak an indigenous language face a mortality risk from WGD that is three times higher than those who do not, while children who receive medical services have a 29% lower risk of mortality compared to those who do not have access to them. It is recommended to offer free medical care in indigenous languages within high-marginalization communities. The combined impact of these policies is expected to significantly reduce child mortality due to WGD.

## 1. Introduction

The Economic Commission for Latin America and the Caribbean (CEPAL in Spanish) reported that in 2019, the crude mortality rate in Mexico was 6.54, while the mortality rate for children under 5 years old was 14.14% [1], which is twice the global rate. This disparity highlights the increased vulnerability of young children and the critical need to examine the factors contributing to these elevated mortality rates.

In that same year, Mexico’s Ministry of Health reported a significant burden of waterborne diseases, with 6 million new cases, indicating a national incidence rate of 4840 per 100,000 inhabitants [2]. This prevalence underscores the severe public health issue posed by inadequate water, sanitation, and hygiene services, with a mortality rate of 3.5 per 100,000 inhabitants attributed to these conditions. Waterborne gastrointestinal diseases (WGD) account for 85% of deaths related to water exposure, highlighting a widespread risk that disproportionately affects the country’s children [3,4].

The statistics describe a troubling situation regarding child morbidity and mortality, with particular concern for school-aged children, considering the immediate and long-term consequences for public health and the well-being of the population. This study aims to address these issues through a detailed analysis of the factors contributing to child mortality from WGD and the formulation of public policies to reduce this burden in the most vulnerable communities.

This research focuses on understanding how various socioeconomic, environmental, cultural, and infrastructural factors influence child mortality due to WGD. The literature review highlights the prevalence of child mortality from WGD in several countries and demonstrates its impact on global public health. Below are key studies from different regions of the world, each offering unique perspectives on this public health issue based on a literature review covering the period from 2005 to 2023 (Table 1).

In Table 1, specialized academic publications addressing child mortality due to WGD are presented, classified by region of study. Each study is detailed with information on the authors, Research Objectives, Methodological Approaches (MA) and Statistical Methods (SM); Statistical Unit (SU) and Key Variables (KV), and Main Findings. These publications significantly increase our understanding of the complex interactions among socioeconomic, environmental, cultural, infrastructural, and behavioral factors influencing this public health problem.

Studies examine a range of factors in explaining diseases and child mortality. Globally, parental education and child mortality have been identified as key areas of focus. For example, the study by Balaj et al. [16], which includes data from 92 countries, analyzes the relationship between parental education and child mortality, showing that higher maternal and paternal education is associated with a significant reduction in child mortality.

Several studies provide a comprehensive view of the relevant variables in investigating child mortality associated with WGD in various socioeconomic contexts, such as individuals, households, communities, countries, and regions of the world. Studies in Latin America highlight the importance of socioeconomic and educational factors, water quality and sanitation, and environmental pollution in water bodies as key determinants in the prevalence of gastrointestinal diseases. These factors are analyzed at different statistical units, including population groups (children, adults), households, and communities (see Table 1). For instance, studies such as those by Zuta Arriola et al. [5] in Peru and Lara Figueroa and García Salazar [7] in Mexico, emphasize socioeconomic factors like poor access to drinking water and sanitation through WASH services, overcrowding in houses, and lack of health education among parents. These studies suggest that deficiencies in sanitation infrastructure and the lack of knowledge or proper hygiene practices significantly contribute to the prevalence of gastrointestinal diseases among children. The study conducted in Argentina by Galiani et al. [8] investigates the impact of water service privatization on child mortality. It concludes that privatization led to a significant reduction in child mortality, especially in poorer areas, with an overall reduction of 8% and an even greater impact (26%) in the most disadvantaged regions. This positive effect can be attributed to improvements in water quality, safety, and the reliability of water supply following privatization, which allowed for better access to clean water and reduce the incidence of waterborne diseases, a major factor contributing to child mortality.

Research in Africa emphasizes deficiencies in water services, storage conditions, sanitation, and hygiene as key determinants of child morbidity and mortality from WGD. Studies such as those by Ohwo and Omidiji [10] and Mebrahtom et al. [11] highlight the influence of WASH services and how unsafe water storage practices and poor sanitation conditions exacerbate the risks of gastrointestinal diseases in children. When water services are inadequate, families often resort to storing water for long periods in unsafe conditions, which increases the likelihood of contamination. These poor storage practices, combined with limited sanitation infrastructure, create environments where gastrointestinal diseases thrive, particularly affecting vulnerable children.

In Asia, climatic conditions and water pollution are emphasized as influential factors. The works of Chen et al. [12] and He & Perloff [13] reveal that environmental factors (e.g., quality of water sources) are crucial in child mortality. In the United States, research focuses on water sources and their connection to acute diarrheal diseases. Studies by Gorelick et al. [15] and Rhoden et al. [14] specifically examine the risk factors associated with water-related illnesses.

The comparative analysis of methodologies used in research on child mortality due to WGD reveals a wide range of approaches and statistical methods (see the third column of Table 1). Researchers have employed statistical methods ranging from descriptive analyses in various geographical contexts to economic and binomial logistic models, each tailored to the specific needs and characteristics of the study.

For instance, in Latin America, studies have utilized cross-sectional descriptive approaches, as seen in Zuta Arriola et al. [5] in Peru, and Lara Figueroa and García Salazar [7] in Mexico, to evaluate disease probabilities (see Table 1). Their statistical methods, including descriptive analysis, chi-square tests., and logistic regression, have enabled researchers investigated the relationships between socioeconomic, environmental, and public health factors. Similarly, in Africa and Asia, research has employed methods such as ANOVA, T-tests, and prospective cohort studies, as demonstrated by Ohwo and Omidiji [10] in Nigeria and Chen et al. [12] in China, respectively (see Table 1). These statistical methods have been essential in understanding disease trends and patterns in varying climatic and sanitary environments.

Globally, the mixed-effects meta-regression used by Balaj et al. [16] exemplifies the application of advanced methodologies to synthesize data from multiple studies, providing a more robust and generalizable analysis of the relationship between parental education and child mortality. In the United States, case-control studies, such as that conducted by Gorelick et al. [15], have been employed to identify specific risk factors associated with water sources and the incidence of diarrheal diseases (see Table 1).

Most of these studies underscore the importance of integrated approaches that address both basic infrastructure and the social determinants of health to improve child health outcomes (see Table 1). Furthermore, the literature review reveals that, in addition to physical and environmental factors, socioeconomic and educational aspects are also critical. These methodologies inform the development of our own research approach by offering valuable insights into the most effective practices and techniques for addressing the complexity of WGD in children.

Although WGD has been identified as a significant determinant of child mortality, accounting for 15% of such deaths in Mexico, existing research primarily focuses on younger cohorts (see Table 1). These approaches are crucial for developing effective intervention policies and programs tailored to the specific needs of younger population. Our study aims to provide a more in-depth and contextualized understanding of this public health issue.

### Objective

The primary goal of this research is to identify and quantify the causes of child mortality due to WGD in school-aged children (3 to 15 years old) in Mexico, a demographic group that has been under-researched in this context. This lack of attention is largely due to the perception that children in this age group, after surpassing the high-risk category of 0 and 3 years, are no longer considered as vulnerable and can integrate into the basic education system with minimal risks.

In this context, the research aims to:Bridge the knowledge gap by assessing WGD-related mortality at the national level.Highlight the prevalence of WGD in high-risk communities with varying socioeconomic conditions.Identify and elucidate the persistent factors contributing to this public health challenge.Propose an effective strategy to address and reduce WGD-related mortality, benefiting Mexican families.

## 2. Materials and Methods

### 2.1. Child Mortality Due to Water-Related Diseases

In Mexico, the National Institute of Statistics and Geography (INEGI in Spanish) reported that in 2019, the total number of deaths was 747,784, of which 88.8% were attributed to health-related diseases, with the remainder due to accidents, homicides, and suicides [17]. The World Health Organization (WHO) reported that in the same year, the mortality rate attributed to unsafe water, sanitation, and hygiene services was 3.5 per 100,000 inhabitants [4], indicating that 4378 people died from water-related diseases. This group of deaths represents 0.58% of total mortality. Of the deaths related to water exposure, over 85% (3721) were due to WGD contracted from consuming or being exposed to contaminated water in Mexico.

WGD were identified by reviewing reports and records on the incidence of waterborne diseases in Mexico by Ministry of the Environment and Natural Resources (SEMARNAT in Spanish) [18], disease catalogs from the Ministry of Health [2], and the International Statistical Classification of Diseases and Related Health Problems provided by the Pan American Health Organization (PAHO) [19].From the list of WGD in Mexico, the research investigates which of these have caused deaths among school-aged children. For this, the death database from the Ministry of Health [3] was reviewed, and cases where children’s causes of death matched the WGD present in the country were identified. The results are presented in Table 2, which lists the eight diseases that are the leading causes of child mortality.

In 2019, a total of 8083 deaths among school-aged children (3 to 15 years) occurred across 3104 localities in the country [3]. Of these, 154 were attributed to the WGD listed in Table 2. The geographic distribution of mortality for both groups of children is depicted in Figure 1.

Figure 1 shows the spatial distribution of child mortality from all causes in Mexico in blue, while mortality attributed to WGD in Mexico is shown in red. The figure illustrates that overall mortality (in blue) is distributed throughout the country, with a higher concentration observed from the central to the southern regions.

The 154 deaths of children caused by WGD (shown in red) occurred in 2019 across 144 communities in 12 states. However, the issue is significantly concentrated in three states: Chiapas, Oaxaca, and Guerrero, which together account for more than four-fifths of the total. Chiapas stands out particularly, accounting for three-quarters of the deaths, followed by Oaxaca with 12% and Guerrero with 7%. This phenomenon is predominantly observed in the localities of these states, which are also considered the poorest in the country.

### 2.2. Methodology

The methodology is established in two stages: first, an analysis of inequality by socioeconomic level in child mortality across the country; and second, the development of an econometric model to determine its causes. This process aims to explore child mortality by employing a combination of inequality measures and econometric models. This holistic approach integrates the Lorenz curve and the Gini coefficient to assess disparities in the distribution of mortality, along with a Generalized Linear Model (GLM) with a logit link function and a binomial distribution.

#### 2.2.1. Measurement of Inequality

To measure the degree of inequality in the distribution of child mortality in Mexico, the database that records the total deaths of school-aged children from all causes is used. This database is sorted according to the Index of Marginalization (IMG) of the localities, which denotes their socioeconomic level according to the National Population Council (CONAPO in Spanish) [20]. IMG is based on indicators related to the communities, including education (illiteracy and lack of basic education), housing conditions (lack of access to water, electricity, or sanitation, and overcrowding), and income level (percentage of the population earning less than two minimum wages). The index rates each community on a gradient ranging from high to very low levels of marginalization. Based on this information, the Lorenz curve is estimated, and the Gini coefficient is calculated, quantifying the inequality in the distribution of child mortality among the different deciles, which reflect 10 groups classified by the socioeconomic level of their locality.

#### 2.2.2. Econometric Model

To investigate the relationship between the risk of mortality due to WGD and the explanatory variables, a Generalized Linear Model (GLM) is used. This model is suitable for handling binary or count responses, using a logit link function and a binomial distribution. The logit link was chosen for its ability to model the relationship between the explanatory variables and the probability of the event of interest, in this case, child mortality due to WGD.

The GLM allows for the incorporation of multiple explanatory variables simultaneously, considering their effect on the dependent variable. The variables were selected based on their theoretical relevance and empirical evidence regarding their association with child mortality due to WGD. This methodology facilitates the estimation of coefficients for each variable and allows for the assessment of the statistical significance of these coefficients, providing a deeper understanding of the factors influencing the risk of child mortality from these diseases in the studied populations.

## 3. Results

### 3.1. Child Mortality from WGD by Socioeconomic Level

The 8083 child deaths that occurred in Mexico from all causes were classified into clusters according to the socioeconomic level of the localities, as defined by the IMG of the CONAPO [20]. The IMG ranges from zero to one, with its value reflecting the socioeconomic conditions of the communities: values close to one indicate poor conditions, while values close to zero indicate very favorable conditions. Thus, deaths of school-aged children were ordered by the IMG value of the locality where they were recorded and classified into 10 groups (deciles) with an equal number of deaths, numbered from I to X. Each decile includes deaths in localities with similar socioeconomic levels. Decile I groups 10% of the deaths in localities with an extremely high level of marginalization, while Decile X includes 10% of deaths in localities with the lowest level of marginalization (see *X*-axis, Figure 2).

Subsequently, the WGD mortality rate for each decile is calculated by assessing the deaths within each group attributed to these diseases. This rate is determined by calculating the proportion of WGD-related deaths in each decile relative to the total across all 10 deciles. The results are plotted on the *Y*-axis of Figure 2.

The results are shown in Figure 2, highlighting that Decile I, which groups localities with an extremely elevated level of marginalization, has a WGD mortality rate of 44%. Decile II records a WGD mortality rate of 15%, and consecutively, the mortality rate decreases dramatically towards the higher deciles, demonstrating an inverse relationship between the socioeconomic level of communities and the risk of death from these diseases. This approach facilitates the identification of significant inequities and underscores the importance of including the socioeconomic context when analyzing disparities in child mortality due to WGD.

#### Inequality in Child Mortality Rate Due to WGD Across Locality Deciles

The inequality in the distribution of child mortality due to WGD in Mexico is measured and illustrated through the Lorenz Curve (Figure 3). This curve is based on the information from Figure 1, but now the values of the X and Y variables represent cumulative data, so Decile X records a value of 100% for both variables. Additionally, in the Lorenz Curve, the *Y*-axis represents the deciles, and the *X*-axis represents the WGD mortality rate.

Figure 3 illustrates the Lorenz Curve depicting the distribution of WGD mortality rates across different deciles. The dotted line represents perfect equity, where each decile accounts for 10% of the WGD mortality rate. In contrast, the solid, concave line shows the actual distribution, revealing significant deviation from equality. The area between these two curves quantifies the inequality among communities with varying socioeconomic conditions. Notably, Deciles I and II, representing localities with extremely high and very high marginalization, account for 59% of total WGD-related deaths, with Decile I alone comprising 44%. This underscores the disproportionate concentration of risk in highly marginalized areas. From Decile III onward, each decile’s share of WGD mortality rate drops below 10%, indicating a decrease yet persistence of mortality across all groups.

The Gini Coefficient of 0.48 indicates a high degree of inequality in child mortality due to WGD. This value suggests an urgent need for targeted interventions in the most disadvantaged regions of Mexico, underscoring a clear relationship between child mortality and the marginalization index.

### 3.2. Causes of Child Mortality

A key finding of this research is the identification of an inverse relationship between the socioeconomic conditions of localities and the WGD mortality rate. However, it is important to investigate which specific community conditions contribute to the presence of WGD, and which variables related to the life history of the children and their families explain why preventable diseases result in premature deaths. To identify the associations between WGD and child mortality, the following model was estimated.

#### 3.2.1. Econometric Model Simulating Child Mortality from WGD in Mexico

The model is specified as a Generalized Linear Model (GLM) with a logit link function and a binomial distribution, which transforms the probability of occurrence into a logit scale, facilitating the modeling of linear relationships between explanatory variables and the probability of mortality. The model simulates the complex relationships between mortality risks and various explanatory variables, revealing the relative importance of socioeconomic determinants and basic services in communities.
(1)$$\log\left(\frac{y}{1-y}\right)=\beta_{0}+\beta_{1}X_{1}+\beta_{2}X_{2}+\beta_{3}X_{3}+\beta_{4}X_{4}+\beta_{5}X_{5}+\beta_{6}X_{6}+\beta_{7}X_{7}$$
where:
The dependent variable y is the percentage of deaths due to WGD out of the total deaths from all causes in each community, interpreted as the probability of mortality in each locality (ranging from 0 to 1).$\left(\frac{y}{1-y}\right)$ is defined as the odds ratio (α), indicating a direct relationship between the probability of the event occurring and the independent variables.$\log\left(\frac{y}{1-y}\right)$ represents the log odds of mortality due to gastrointestinal diseases.$\beta_{0}$ denotes the model intercept.$\beta_{1},\beta_{2},\ldots \beta_{7}$ are the model coefficients that reflect the impact of each independent variable.$X_{1},X_{2},\ldots ,X_{7}$ are the independent variables incorporated into the model.


#### 3.2.2. Model Variables

The model relates the incidence rate of mortality from WGD (the dependent variable) to seven explanatory variables (independent variables), simulating the complex relationships between mortality risks and these variables. Table 3 describes the variables included in the model and their characteristics.

Table 3 presents seven independent variables, classified into two groups. The first group includes three variables that describe individual characteristics of the deceased children: their age, whether they speak an Indigenous language, and whether they are beneficiaries of a health program. The second group includes community characteristics where the incident occurred: the locality’s IMG, and the percentages of the population without access to piped water, living in overcrowded conditions, (having more than 2.5 persons per room) and without basic education (referring to individuals over 15 years old who have not completed primary and/or secondary education in Mexico).

The data used to estimate the model were compiled as follows. Community characteristics were obtained from the IMG database by locality from the National Population Council [19]. Information on the children’s characteristics was extracted from the 2019 Mexican death records database from the Ministry of Health [3]. This information was integrated into a single database using the locality marginalization index as a linking variable.

A GLM is characterized by its flexibility in handling binary outcome variables and its ability to model the probability of an event occurring, making it particularly suitable for this analysis. It allows for the examination of the relationship between a set of predictive factors and mortality probability, providing a detailed understanding of the impact of socioeconomic and environmental determinants on child health in communities. The selection of variables and the use of the GLM with a logit link function in this study are strategically designed to address the multifaceted nature of child mortality due to WGD.

#### 3.2.3. Model Estimation Results

The model was estimated using Stata/MP 17 software, and the results are described in this section. Table 4 presents the statistical values that demonstrate the goodness-of-fit and overall effectiveness of the model.

Model Fit Evaluation. Key statistical results indicate that the GLM has a good fit and is effective in modeling the probability of child mortality due to WGD in the communities analyzed. The deviance per degree of freedom and the Pearson statistic per degree of freedom are both less than 1, suggesting an adequate fit. Additionally, the low and negative values of AIC and BIC reinforce the robustness of the model. This analysis indicates that the model is well-suited to capturing the complex relationships between explanatory variables and the probability of child mortality from WGD, providing a solid foundation for interpretation and policy formulation. Below is a detailed interpretation of the model results.

Deviance per Degree of Freedom and Pearson Statistic:

Deviance and the Pearson statistic are measures of model goodness-of-fit. Values close to 1 for deviance per degree of freedom and the Pearson statistic per degree of freedom indicate a good model fit. The deviance per degree of freedom (0.0742824) and the Pearson statistic per degree of freedom (0.129254) are both less than 1, suggesting that the model has an adequate fit, as values less than or close to 1 indicate that the model fits the data well.

Information Criteria (AIC and BIC):

Low AIC and BIC values indicate a good model fit, penalizing for model complexity. The negative AIC and BIC values suggest that the model achieves a good balance between fit and simplicity. The low and negative values of AIC (0.0843448) and BIC (−56,708.76) indicate that the model fits well. These criteria penalize model complexity, suggesting that the model is parsimonious and fits the data well without overfitting.

Log-Likelihood:

The log-likelihood value measures how well the model predicts the observed data. A higher (less negative) value indicates a better fit. The log-likelihood value of −266.9640883 is relatively high (less negative), indicating that the model predicts the observed data well.

#### 3.2.4. Statistical Significance of Parameters and Explanatory Power of Variables

Table 5 presents the estimated model coefficients along with their associated statistics, enabling the assessment of their statistical significance.

Table 5 shows that, except for the variable portraying whether the individual is a beneficiary of medical services, all coefficients are statistically significant at the 95% level. The community conditions variables are statistically significant, indicating their value in explaining the child mortality rate. The variables describing the children’s age and whether they speak an indigenous language are also significantly relevant.

Child’s Age

The age variable has a coefficient of −0.9270412, indicating that a 1% increase in log-age reduces the probability of mortality due to WGD by approximately 0.924%. This finding highlights that younger children are more vulnerable, and as they grow older, their risk of mortality due to WGD decreases slightly.

Speaking an Indigenous Language

The parameter for the variable speaking an indigenous language is statistically significant (*p* < 0.01), indicating that this cultural characteristic has a significant qualitative impact on improving the health and well-being of communities. As a dichotomous variable, the coefficient of 1.121947 is associated with a significant increase in the log-odds: the odds of mortality due to WGD are 3.07 times higher for a child who speaks an indigenous language compared to one who does not, holding all other variables constant. It underscores the need for targeted interventions to overcome cultural and linguistic barriers in healthcare to reduce child mortality due to WGD.

Access to Health Services

In the model, the parameter for being a beneficiary of a health program is not statistically significant (*p* > 0.05). The variable is dichotomous, and its coefficient of −0.3392942 suggests a reduction in the log-odds of mortality due to WGD for children who are beneficiaries of medical services. This means that the probability of mortality in this group is reduced by approximately 28.8% compared to those who are not beneficiaries. This result suggests the need to improve the quality and effectiveness of access to medical services to achieve a significant impact on reducing WGD mortality. It also suggests evaluating and enhancing the quality and effectiveness of health programs to ensure they truly contribute to reducing WGD mortality in the child population.

Marginalization Index

The socioeconomic conditions of the localities (IMG) have a significant impact on the incidence of child mortality from WGD. The high coefficient value (5.546809) confirms the findings from the inequity indicators discussed above. It indicates that a 1% increase in the IMG is associated with a 5.7% increase in the probability of child mortality.

Population Without Basic Education

The coefficient for the proportion of the population without basic education is 2.14, indicating that a 1% increase in this variable corresponds to an approximate 2.2% rise in the probability of mortality. This finding underscores the critical importance of improving basic education as a key strategy to reduce child mortality.

Population Living in Overcrowded Housing

The percentage of occupants in overcrowded housing shows a coefficient of 1.62, suggesting that a 1% increase in overcrowding raises the probability of mortality due to WGD by approximately 1.6%. This finding emphasizes the need to improve housing conditions to reduce child mortality.

Population Without Access to Piped Water to their homes or premises

The proportion of the population without access to piped water has a coefficient of 0.238, which means that a 1% increase raises the probability of mortality due to WGD by approximately 0.24%. This finding underscores the critical importance of improving access to potable water.

Summary of Findings

The variables representing community characteristics are statistically significant, positively correlated with mortality, and have a substantial impact on the likelihood of death from WGD. These factors highlight the need to improve education, access to potable water, and housing conditions, as well as to address socioeconomic marginalization to reduce child mortality across communities. Targeted interventions in these areas could be crucial for improving child health.

### 3.3. Strategy to Reduce Child Mortality Due to WGD

The model’s goodness-of-fit and the significance of its parameters form the foundation for designing a strategy aimed at reducing the child mortality rate from WGD by 1.9%. This strategy involves defining policies based on five variables from the model, which are applied simultaneously. The policy design assesses the varying capacities of these variables to influence the mortality rate and evaluates their feasibility for implementation in the affected communities.

Table 6 provides a summary of the strategy: the first column lists the model variables included in the strategy; the second column outlines the public policies and the magnitude of change; the third column describes the necessary interventions to mitigate mortality and improve child health; and the fourth column details the expected impact of each policy on reducing the child mortality rate due to WGD.

The strategy outlined in Table 6 assumes that the concurrent implementation of these policies will significantly reduce the mortality rate due to WGD in Mexico. Achieving this goal depends on the simultaneous application of all proposed interventions targeting key The strategy involves the concurrent implementation of the following policies: increasing the coverage of access to piped water by 2.51%; reducing the national average IMG with a focus on localities with extreme marginalization; improving average housing conditions by 0.25 points; increasing the percentage of adults with basic education by 0.25 points; and finally, increasing the life expectancy of school-aged children by 0.1 percentage points through proper care from an early age. These policies are designed to have direct and indirect impacts on WGD mortality, providing a holistic approach to improving child health in marginalized communities.

The central policy in the strategy focuses on improving the provision of piped water, aiming to increase the percentage of the population with access to piped water in affected communities by 2.51%. Expanding service coverage is expected to reduce the national mortality rate by 0.6%. The expansion of potable water infrastructure and the implementation of education programs on hygiene and sanitation practices are essential to achieving this goal. The model highlights that ensuring better water service to households is crucial to preventing water-related diseases. The scenario presented in the strategy for reducing child mortality anticipates that by increasing piped water coverage by 2.51%, the national mortality rate could be reduced by 0.6%, which equates to preventing 49 child deaths out of a total of 8083 deaths from all causes. This reduction is particularly significant in communities most affected by WGD. Although the policy of reducing the population without access to piped water by 2.51% may seem costly in terms of initial infrastructure investment, it is essential to highlight that the cost of implementing this policy will always be lower than the value of saving the lives of 49 children. Moreover, the collateral benefits of improving access to potable water are vast and significant. For example, by reducing the prevalence of WGD, not only will short-term child mortality be avoided, but there will also be an additional reduction in deaths in the medium term, both among children and adults. These indirect benefits, such as the overall improvement in public health and the reduction of medical costs associated with these diseases, more than compensate for the initial infrastructure investment. Among the set of policies analyzed, improving access to piped water is the most promising, given its immediate impact and its potential to generate positive secondary effects on public health. However, to definitively support this conclusion, it would be necessary to conduct a more detailed cost-benefit analysis, which would allow for a precise quantification of the net benefits derived from this intervention. The second most important policy focuses on increasing the education level of the adult population. This policy is overly sensitive to changes; even a 0.23% increase in the percentage of the population with basic education reduces the mortality rate by 0.5%. These improvements can be achieved through improvements in access to and quality of basic education, adult literacy programs, and awareness campaigns about the importance of education.

The third key policy focuses on reducing the percentage of the population living in overcrowded conditions. This policy is overly sensitive; a 0.25-point reduction decreases the mortality rate by 0.4%. This reduction can be achieved through improvements in housing conditions, the construction of adequate housing, subsidy programs to improve family living conditions, and sustainable urban planning.

The fourth policy focuses on improving the socioeconomic conditions of localities across the country, as reflected in the IMG, which, as expected, is the most sensitive variable in the model. The strategy aims to achieve a modest reduction in the national average IMG (0.05%), which reduces the child mortality rate by 0.3%. The proposal is to focus on communities, with additional interventions alongside the three policies mentioned above. These interventions include improving community infrastructure (such as roads, markets, recreational spaces, hospitals), increasing educational opportunities with well-built schools, creating economic development programs, and implementing cultural and anti-discrimination measures.

Finally, the fifth policy in the strategy focuses on increasing children’s life expectancy, a variable that exhibits unitary elasticity. This implies that a 0.11% increase in children’s life expectancy results in an equivalent reduction (0.11%) in the child mortality rate. To achieve this, interventions should include implementing comprehensive child healthcare and monitoring programs, ensuring the availability and accessibility of vaccines, and providing appropriate medical care from birth through school age.

#### Importance of Dichotomous Variables in the Strategy

Although the dichotomous variables do not have statistical significance in directly reducing the WGD mortality rate, their qualitative impact on the probability of child mortality is significant. A notable finding is that a child who speaks an indigenous language has a 3.07 times higher probability of mortality compared to a child who speaks Spanish. This result underscores the importance of providing medical care with personnel who speak the same indigenous language and implementing policies that improve the quality and accessibility of healthcare to mitigate the disadvantages faced by these children.

## 4. Discussion

The research methodology employed in this study is structured in two distinct stages. The first stage focuses on highlighting the profound social and economic inequalities reflected in the mortality rates due to WGD. This stage involves a detailed analysis of child mortality across the country, stratified by socioeconomic levels, to uncover the disparities in mortality rates among different socioeconomic groups.

The second stage involves the development of an econometric model to identify and quantify the underlying causes of these disparities. The comprehensive nature of the data, which covers the entire national territory and encompasses a wide range of socioeconomic and geographical variations, ensures that there are no limitations concerning data quality or representativeness, thereby enhancing the validity and reliability of the study’s findings.

The results from the first stage of analysis reveal significant disparities in child mortality rates associated with WGD, underscoring a clear correlation between these mortality rates and the marginalization index of localities. Localities with an extremely high level of marginalization are disproportionately affected, accounting for 44% of child deaths, while those with a very high level of marginalization represent an additional 15% of deaths. These disparities are quantitatively expressed through an elevated Gini coefficient of 0.48, which surpasses the national income inequality index (Gini coefficient of 0.41), thereby highlighting the deep-rooted social and economic inequalities that contribute to differential mortality outcomes.

In the second stage, a Generalized Linear Model (GLM) with a logit link function is employed to analyze the complex interactions among seven variables that influence child mortality. The results demonstrate that community-level characteristics—specifically, the percentages of the population lacking piped water, experiencing overcrowded living conditions, and having limited access to education—are positively correlated with higher probabilities of mortality and are statistically significant. The marginalization index (IMG) emerges as a statistically significant variable with an inverse and highly sensitive relationship to the mortality rate, corroborating the patterns observed in the first stage. Additionally, variables describing individual child characteristics, such as language spoken and access to medical services, show varied impacts: a child who speaks an indigenous language (a statistically significant factor) has a 3.07 times greater likelihood of mortality from WGD compared to a child who speaks Spanish. Furthermore, a child not covered by medical services would see their probability of mortality decrease by approximately 29% if they were to gain access to such services, although this finding was not statistically significant. The age variable, which is statistically significant, serves as a proxy for child life expectancy and directly impacts mortality probability. Further exploration reveals that indigenous language speakers are often concentrated in households with disproportionately lower rates of piped water and higher levels of overcrowding, suggesting that cultural factors are intertwined with socioeconomic disadvantages. This correlation between culture, living conditions, and access to basic services may further exacerbate health disparities. Moreover, marginalized locations where indigenous populations reside tend to have higher rates of deprivation in multiple dimensions, including access to clean water and adequate housing, contributing to their vulnerability to WGD.

The application of the GLM in this study offers distinct advantages, particularly in its ability to handle binary dependent variables while incorporating both quantitative and qualitative factors. This methodological approach aligns with research conducted in Mexico by Lara-Figueroa and García-Salazar [7] and Rodríguez-Tapia and Morales-Novelo [9], as well as in Argentina by Galiani et al. [8]. Unlike simpler methods, such as the descriptive and cross-sectional approach used by Zuta-Arriola et al. [5] or the binomial models employed by Lara-Figueroa and García-Salazar [7], the GLM facilitates a more nuanced analysis of the interactions between individual factors (e.g., speaking an indigenous language) and community-level factors (e.g., access to basic services) in shaping health outcomes. This flexibility is crucial in public health research, where data rarely conform to ideal distributions, and socioeconomic and cultural conditions can vary widely.

The use of the GLM also facilitates a deeper interpretation of results, allowing policymakers to better understand how changes in explanatory variables affect mortality probabilities. For instance, while the variables of access to medical services and speaking an indigenous language did not show direct statistical significance in reducing mortality rates, their qualitative impact—revealed through the GLM—underscores the need for culturally inclusive health services. This approach not only supports the identification of priority interventions but also enhances the communication of these findings to policymakers.

Comparing the findings of this study with international research on child mortality due to WGD reveals both similarities and differences across diverse socioeconomic and geographical contexts. Studies from Peru and Brazil [5,6] have similarly identified socioeconomic factors, such as overcrowding and access to clean water, as significant determinants of child health outcomes. In Argentina, the study by Galiani et al. [8] demonstrated that water service privatization contributed to reducing child mortality, especially in poorer areas. However, this strategy might not be directly applicable to Mexico, where public water infrastructure and cultural factors, such as language barriers, present more prominent challenges. In Africa and Asia, research underscores the role of WASH (Water, Sanitation, and Hygiene) services in mitigating diarrheal diseases, as evidenced by studies from Ethiopia [11] and Nigeria [10].These findings align with the current study’s emphasis on improving access to potable water and sanitation services in marginalized communities.

Globally, the influence of parental education on child mortality is well-documented, as highlighted by Balaj et al. [16]. The present study confirms this, showing that improved parental education is associated with better child health outcomes in Mexico. However, additional variables unique to the Mexican context, such as speaking an indigenous language and access to culturally inclusive medical services, are less explored in other regions but play a significant role in shaping health outcomes in Mexico. This underscores the need for context-specific interventions that address both universal determinants, such as education and water access, and localized factors, such as cultural and linguistic barriers.

The results also highlight the severe disparities in child mortality, particularly in communities with extreme and very high levels of marginalization. This suggests that addressing child mortality requires not only health interventions but also broader social and economic reforms.

The findings of this study have direct implications for public policy in Mexico. Targeted interventions focusing on increasing access to potable water, improving education, and considering the inclusion of information about access to schools in Mexico—both for the education of children and parents, and for the water services available in schools —could provide a deeper understanding of how these factors influence public health outcomes. Moreover, addressing overcrowded living conditions are essential, particularly in highly marginalized communities. Additionally, the study highlights the importance of culturally inclusive strategies, especially in regions where indigenous languages are spoken, to ensure that health services are accessible to the entire population.

The methodology applied in this research is ideal for regions with high socioeconomic differentiation. This process aims to explore child mortality by employing a combination of inequality measures and econometric models. Despite the advantages of the GLM, the study’s limitations must be acknowledged. The cross-sectional nature of the data restricts the ability to capture temporal dynamics, suggesting that future research could benefit from incorporating longitudinal data. Furthermore, the inclusion of spatial analyses could provide greater precision in identifying geographical patterns that influence child mortality, allowing for more effective targeting of resources and interventions.

## 5. Conclusions

Child mortality in Mexico due to WGD remains a critical public health challenge, especially in the country’s most marginalized communities. In 2019, 8083 deaths were recorded among children aged 3 to 15 years, with 1.9% of these deaths attributed to WGD—deaths that could have been prevented with appropriate interventions.

This study reveals stark inequalities in the distribution of child mortality related to WGD, disproportionately affecting localities with extreme and very high levels of marginalization, especially in the states of Chiapas, Oaxaca, and Guerrero. The econometric model used in this research identified seven key factors influencing child mortality, encompassing both individual characteristics (e.g., age, access to medical services, and speaking an indigenous language) and community-level factors (e.g., access to potable water, parental education, housing conditions, and the level of marginalization). The findings underscore the significant impact of socioeconomic determinants, particularly access to potable water, education, and improved living conditions, on reducing child mortality.

Although access to medical services and speaking an indigenous language—both dichotomous variables—did not directly reduce mortality rates, they play a critical qualitative role in mitigating mortality risks among disadvantaged children. This highlights the importance of culturally inclusive health services and the need to address linguistic barriers to ensure equitable access to healthcare. Providing culturally sensitive care and overcoming language barriers are essential to improving health outcomes for indigenous populations, particularly in regions where indigenous languages are predominant.

Based on these findings, the policy recommendations are clear: improving water infrastructure, enhancing parental educational opportunities, and addressing overcrowded living conditions should be prioritized in regions of high marginalization. However, it is crucial to emphasize that the policy aimed at increasing the percentage of the population with access to piped water must be applied in all localities with deficits in this service, as the lack of potable water is clearly related to child mortality due to WGD. Additionally, implementing culturally and linguistically inclusive health services, particularly for indigenous populations, is essential for effectively reducing child mortality.

Finally, this study provides a foundation for evidence-based public health policies aimed at reducing child mortality due to WGD in Mexico. To further refine these strategies, future research should incorporate spatial and longitudinal analyses to gain a deeper understanding of the geographic and temporal dimensions of this issue. Continuous research and adaptive policy measures will be crucial to achieving long-term reductions in child mortality and addressing the broader social and economic inequalities that contribute to these outcomes.

## Figures and Tables

**Figure 1 ijerph-21-01399-f001:**
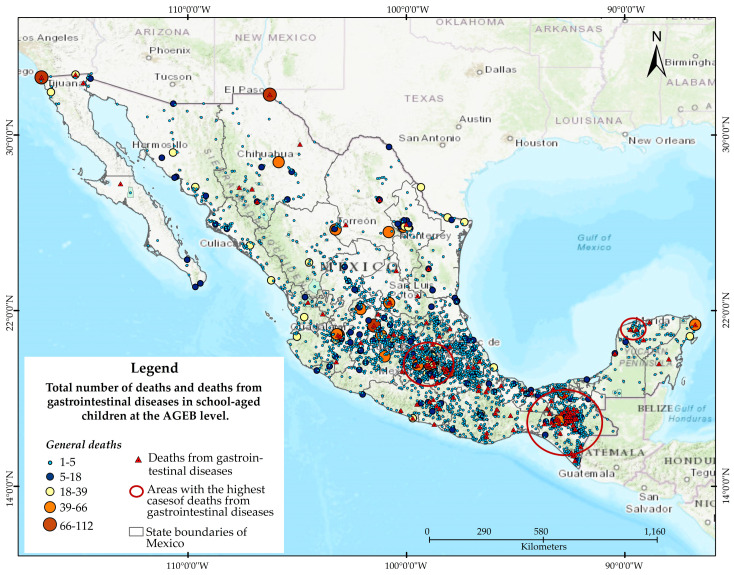
Child mortality from WGD in Mexico (2019). Source: Own elaboration based on Ministry of Health [3] data, with spatial analysis conducted using ArcGIS 10.8.

**Figure 2 ijerph-21-01399-f002:**
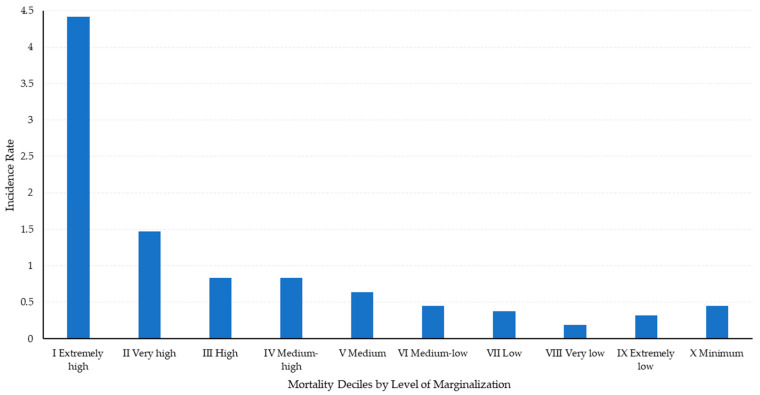
Child Mortality from WGD by Socioeconomic Level. Note: The Level of Marginalization (NM) is defined based on the IMG of each decile. The NM denotes the socioeconomic conditions of the localities included in each decile.

**Figure 3 ijerph-21-01399-f003:**
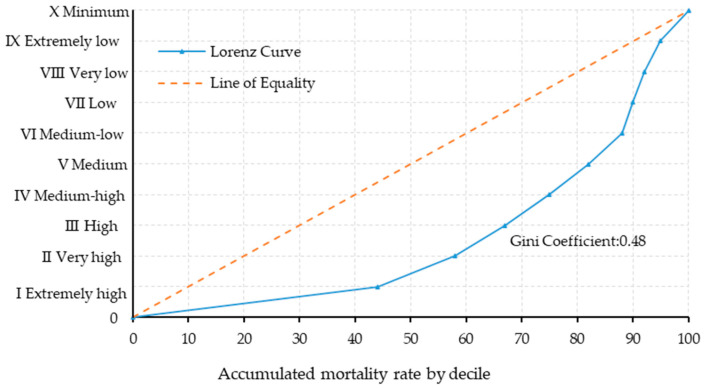
Lorenz Curve Relating Child Mortality Rate by WGD and Socioeconomic Groups in Mexico.

**Table 1 ijerph-21-01399-t001:** Publications on Waterborne Gastrointestinal Diseases (WGD) and Child Mortality (2005–2023 Literature Review).

Study by Region	Research Objectives	Methodological Approaches (MA) and Statistical Methods (SM)	Statistical Unit (SU) and Key Variables (KV)	Main Findings
**Latin America**				
Zuta Arriola et al. 2019 [5]— **Perú**	Evaluate the impact of health education programs on gastrointestinal disease prevalence among preschool children, considering socioeconomic factors like household conditions and access to sanitation.	MA: Cross-sectional descriptive study.	SU: Households and preschool children.	The health education program reduced gastrointestinal disease prevalence and improved sanitation, water quality, and hygiene. Overcrowding was significantly associated with parasitosis.
		SM: Descriptive analysis and chi-square tests.	KV: Overcrowding, household size, sanitation, presence of animals.	
Souza et al. 2014 [6]— **Brasil**	Analyze the relationship between sanitary quality and natural disasters with morbidity and hospitalizations due to WGD in children under five.	MA: Panel study.	SU: Municipalities and children under five.	A 1% improvement in sanitary quality reduced hospital bed occupancy by 1.02%, highlighting the role of sanitation in public health. Natural disasters had a significant but smaller impact on morbidity.
		SM: Non-linear econometric model using panel data.	KV: Sanitation, water quality, natural disasters.	
Lara Figueroa and García Salazar, 2019 [7]—**México**	Determine the prevalence of WGD related to contaminated water use in households lacking potable water and sanitation in the Valle del Mezquital.	MA: Cross-sectional descriptive study.	SU: Households and individuals with gastrointestinal diseases.	The lack of basic services and access to potable water significantly increased the likelihood of gastrointestinal diseases in households. Households with piped water and proper sanitation had lower risks.
		SM: Logistic regression and chi-square tests.	KV: Potable water access, sanitation, sociodemographic variables.	
Galiani et al., 2008 [8]— **Argentina**	Identify the effect of water privatization on child mortality rates in municipalities that privatized their water supply systems.	MA: Quasi-experimental study.	SU: Municipalities and child mortality (under five).	Water privatization was associated with a reduction in child mortality by 8%, with a more pronounced effect of 26% in poorer regions. This reduction was driven by improved water infrastructure and increased access to water services in areas that privatized their water systems.
		SM: Differences-in-differences analysis.	KV: Water service ownership, child mortality, socioeconomic data.	
Rodríguez Tapia and Morales Novelo, 2017 [9]—**México**	Determine the role of Atoyac River contamination in contributing to gastrointestinal disease incidence.	MA: Ecological study.	SU: Communities near the river, water quality stations.	The study found a positive causal relationship between coliform bacteria contamination in the Atoyac River and gastrointestinal diseases in nearby communities.
		SM: Logistic regression model.	KV: Water contamination, gastrointestinal deseases, socioeconomic conditions.	
**Africa**				
Ohwo and Omidiji, 2021 [10]— **Nigeria**	Analyze the prevalence of diarrhea and typhoid fever in relation to WASH services in Yenagoa.	MA: Cross-sectional descriptive study.	SU: Households and children under five.	Inadequate WASH services were linked to higher rates of diarrhea and typhoid fever, especially among children under five, with more cases during the dry season.
		SM: Descriptive statistics and chi-square tests.	KV: WASH status, diarrhea, typhoid fever, demographic factors.	
Mebrahtom et al., 2022 [11]— **Etiopía**	Identify the risk factors contributing to diarrhea-related child mortality in eastern Ethiopia.	MA: Nested case-control study.	SU: Households and children with diarrhea.	Risk factors for diarrhea-related child mortality included unsafe water storage, lack of water treatment, poor sanitation, and improper waste disposal.
		SM: Conditional logistic regression. Bivariate and multivariable analyses.	KV: Water storage, sanitation, child mortality.	
**Asia**				
Chen et al., 2005 [12]— **China**	Evaluate the risks associated with different drinking water sources for colorectal cancer in Jiashan County, China.	MA: Cohort study.	SU: Individuals and drinking water sources.	Colorectal cancer incidence increased with water contamination from various sources. Well water showed the highest risk, followed by mixed water, ditches, rivers, and municipal water.
		SM: Cox proportional hazards regression.	KV: Water contamination, colorectal cancer, water sources.	
He and Perloff, 2016 [13]— **China**	Estimate the effects of surface water quality on infant mortality rates in China.	MA: Ecological study.	SU: Regions in China, infant mortality.	Surface water quality had a non-monotonic relationship with infant mortality. Mortality rates were highest in areas with “acceptable” water quality, as people did not perceive its decline. Worse pollution led to lower water use, reducing mortality.
		SM: Ordered probit model.	KV: Surface water quality, precipitation, socioeconomic factors.	
**North America**				
Rhoden et al., 2021 [14]— **United States**	Determine the prevalence and risk factors for waterborne diseases, water-based toxins, and vector-borne diseases in Florida.	MA: Retrospective descriptive study.	SU: Reported cases of water-related diseases.	Over 218,000 cases of water-related diseases were reported in Florida, mostly waterborne. Children aged 0–4 and the elderly were most affected. The southeast region had the highest incidence, with salmonellosis as the most prevalent pathogen.
		SM: Descriptive statistics and regression analysis.	KV: Waterborne diseases, vector-borne diseases, water toxins.	
Gorelick et al., 2011 [15]— **United States**	Examine the association between water exposures and acute diarrheal illness in children in a U.S. metropolitan area.	MA: Nested case-control study.	SU: Children with diarrheal illness.	Well-water use was associated with a higher likelihood of acute diarrheal illness (ADI) in children compared to surface water. Bottled water did not reduce the risk of ADI for well-water users and increased the odds of ADI for surface water users.
		SM: Conditional logistic regression, ANOVA, and chi-square tests to compare variables.	KV: Water source, drinking water type, water filters.	
**Global**				
Balaj et al., 2021 [16]— **Global**	Estimate the influence of parental education on under-5 mortality reductions, focusing on maternal and paternal education during different age intervals.	MA: Systematic review and meta-analysis.	SU: Children under five years.	Parental education was significantly associated with reduced under-5 mortality. Maternal education had a stronger effect, reducing mortality by 31% for mothers with 12 years of schooling, compared to 17.3% for fathers.
		SM: Mixed-effects meta-regression models and meta-analysis using multiple datasets.	KV: Parental education, under-five mortality, wealth.	

Source: Prepared by the authors based on an extensive review of relevant literature 2005–2023.

**Table 2 ijerph-21-01399-t002:** Waterborne Gastrointestinal Diseases in Mexico that cause child mortality.

Intestinal Amebiasis (A06.0–A06.3, A06.9)
Cholera (A00)
Diarrhea and Gastroenteritis of Presumed Infectious Origin (01H) *
Paratyphoid Fever A (A01.1)
Typhoid Fever (A01.0)
Intestinal Infections by Other Organisms and Poorly Defined (A04, A08-A09 except A08.0)
Food Poisoning (01E) *
Shigellosis (01D) *

Sources: Own elaboration based on information from the Ministry of Health [3], PAHO [19] and * Ministry of Health. [3]. Note: The codes in parentheses are based on the International Statistical Classification of Diseases and Related Health Problems (ICD-10). Asterisks (*) indicate codes that were not provided directly and are inferred based on the context.

**Table 3 ijerph-21-01399-t003:** Variables included in the Model.

Variable	Description	Original Coding	Modified
Dependent:			
y	Children who died from WGD relative to the total number of child deaths in the locality (ages 3–15)	Percentage	
Independent:			
Child Characteristics			
$X_{1}$	Indigenous language speaker	Dichotomous No = 0, Yes = 1	
$X_{2}$	Age of school-aged children	Years	Logarithm
$X_{3}$	Beneficiary of medical services	Dichotomous Yes = 1, No = 0	
Community Characteristics			
$X_{4}$	Locality marginalization index (IMG), 2020	Percentage	
$X_{5}$	Residents in households without piped water to their home or premises.	Percentage	Logarithm
$X_{6}$	Residents living in overcrowded conditions households	Percentage	Logarithm
$X_{7}$	Population aged 15 and older without basic education	Percentage	Logarithm

**Table 4 ijerph-21-01399-t004:** Model Goodness-of-Fit Statistics.

Generalized Linear Model	Number of Obs = 6520
Optimization: ML	Residual df = 6512
Scale parameter = 1	Deviance = 483.7268779
(1/df) Deviance = 0.0742824	Pearson = 3991.370253
(1/df) Pearson = 0.129254	
Variance function: V(U) = U * (1 − U/1) [Binomial]	
Link function: g(u) = ln(u/(1 − u)) [Logit]	
AIC = 0.0843448 BIC = −56,708.76	Log-likelihood = −266.9640883

**Table 5 ijerph-21-01399-t005:** Coefficients and Significance.

Variables	Notation	Coefficient	Std. Err.	Z Value	*p* > z	95% Confidence	
Interval							
Constant	β0	−2.605668	1.669129	−1.56	0.119	−5.8771	0.6657645
Indigenous language speaker	β1	1.121947 *	0.3468375	3.23	0.001 *	0.4421577	1.801736
Age of school-aged children	β2	−0.9270412 *	0.2325879	−3.99	0.01 *	−1.382905	−0.4711773
Beneficiary of medical services	β3	−0.3392942	0.295422	−1.15	0.251	−0.9183106	0.2397222
Locality marginalization index	β4	5.546809 *	2.819234	1.97	0.049 *	0.0212125	11.07241
Residents in households without piped water to their home or premises.	β5	0.2386455 *	0.0943122	2.53	0.011 *	0.0537969	0.4234941
Residents living in overcrowded conditions households	β6	1.627555 *	0.6198872	2.63	0.009 *	0.4125986	2.842512

Note: An asterisk (*) marks values that are statistically significant at the 95% confidence level.

**Table 6 ijerph-21-01399-t006:** Strategies to Reduce the Child Mortality Rate in Mexico.

Variable	Proposed Policy (Variable Change)	Suggested Interventions	Impact on Mortality (%/Number of Deaths Prevented)
Population without access to piped water	Reduce the proportion of the population without access to piped water by 2.51%	Improve access to potable water through the expansion of water supply infrastructure; implement water treatment and purification programs; promote education in hygiene and sanitation practices.	0.6% reduction (49 deaths)
Population without basic education	Reduce the proportion of the population without basic education by 0.23%	Improve access to and quality of basic education; promote adult literacy programs and awareness campaigns on the importance of education; integrate health education into the school curriculum.	0.5% reduction (40 deaths)
Residents in overcrowded households	Reduce the proportion of overcrowding by 0.25%	Improve housing conditions by constructing adequate housing; implement subsidy programs to improve families’ living conditions; promote sustainable urban planning.	0.4% reduction (32 deaths)
Marginalization index	Reduce the marginalization index by 0.05%	Implement policies and programs to reduce socioeconomic marginalization; improve community infrastructure, access to basic services such as potable water and sanitation and increase educational and economic opportunities in marginalized communities.	0.3% reduction (24 deaths)
Child’s Age	Increase in age by 0.11%	Implement child-specific health care and monitoring programs for younger children; ensure availability and access to vaccines and proper medical care from birth to school age.	0.1% reduction (8 deaths)
Total impact			1.9% reduction (153 deaths)

## Data Availability

The original contributions presented in the study are included in the article, further inquiries can be directed to the corresponding author/s.

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
