# Peer review of "Waterborne Gastrointestinal Diseases and Child Mortality: A Study of Socioeconomic Inequality in Mexico"

_ijerph, 2024, doi:10.3390/ijerph21111399_

Round 1
Reviewer 1 Report
Comments and Suggestions for Authors
I read the article titled “Waterborne Gastrointestinal Diseases and Child Mortality: A Study of Socioeconomic Inequality in Mexico” by Morales et al. with great interest. Overall, the manuscript flows smoothly, but there are some aspects that, in my opinion, need extensive revision.
In Table 1, you mentioned several studies. The study type column needs reevaluation because t-tests and regression are not study types. Would it be a good idea to briefly mention the results for each study instead of focusing on key variables and research objectives?
Lines 61-66: In its current form, I do not believe the table is helpful for readers.
The introduction is over 1,500 words. I believe an introduction should be around 500 to at most 1,000 words to keep readers engaged. Much of the information you provided is general and could be reorganized into a table.
I would like to know how the authors interpret their results in Table 6, particularly the 0.6% reduction in mortality. Considering the total number of mortalities (around 150), do you think these interventions are cost-effective? Or should we invest in other aspects of disease prevention?
Author Response
|
Comments 1: [In Table 1, you mentioned several studies. The study type column needs reevaluation because t-tests and regression are not study types. Would it be a good idea to briefly mention the results for each study instead of focusing on key variables and research objectives? Lines 61-66: In its current form, I do not believe the table is helpful for readers.]
|
|
Response 1: We agree with this comment. [We greatly appreciate your valuable observation regarding the Study Type column in Table 1. We agree that t-tests and regression are statistical methods and not study types. Consequently, we have revised the classification in this column to accurately reflect the study types used in the mentioned articles, such as descriptive, quasi-experimental, cohort, or case-control studies. Additionally, in response to your suggestion, we have integrated concise summaries of the main findings for each study instead of focusing solely on key variables and research objectives. For greater clarity, the Study Type column has been renamed and revised as Methodological Approaches (MA). This was combined with the Statistical Methods (SM) in a single column to provide a clearer explanation of the methodologies applied. These changes can be found in the revised Table 1, located from lines 61-66 in the manuscript. A new version of Table 1 has been attached to reflect these adjustments. We believe that these modifications enhance the clarity and usefulness of the table, providing readers with a more comprehensive understanding of the study outcomes and methodologies.
|
|
Comments 2: [The introduction is over 1,500 words. I believe an introduction should be around 500 to at most 1,000 words to keep readers engaged. Much of the information you provided is general and could be reorganized into a table.] |
|
Response 2: We appreciate your feedback regarding the length of the introduction. While we understand the concern about keeping readers engaged, we believe that the current length is necessary to provide a comprehensive overview of the complex public health challenges related to waterborne gastrointestinal diseases (WGD) and child mortality. The introduction not only outlines the significance of these issues in Mexico but also allows for a valuable comparison of approaches and statistical methods used in different regions globally. This contrast helps to contextualize the unique factors contributing to child mortality in various parts of the world. The extended introduction enables us to highlight differences in methodologies and the socio-economic conditions that exacerbate child mortality. While we understand your suggestion to move some of this content into a table, we believe that this detailed narrative is essential to set the foundation for the analysis presented in the subsequent sections. Additionally, although the table provides a structured comparison of studies, the introduction lays the groundwork for understanding the broader context and significance of these comparisons. Comments 3: I would like to know how the authors interpret their results in Table 6, particularly the 0.6% reduction in mortality. Considering the total number of mortalities (around 150), do you think these interventions are cost-effective? Or should we invest in other aspects of disease prevention? Response 3: The scenario presented in the strategy for reducing child mortality anticipates that by increasing piped water coverage by 2.51%, the national mortality rate could be reduced by 0.6%, which equates to preventing 49 child deaths out of a total of 8,083 deaths from all causes. This reduction is particularly significant in communities most affected by waterborne gastrointestinal diseases (WGD). Although the policy of reducing the population without access to piped water by 2.51% may seem costly in terms of initial infrastructure investment, it is essential to highlight that the cost of implementing this policy will always be lower than the value of saving the lives of 49 children. Moreover, the collateral benefits of improving access to potable water are vast and significant. For example, by reducing the prevalence of WGD, not only will short-term child mortality be avoided, but there will also be an additional reduction in deaths in the medium term, both among children and adults. These indirect benefits, such as the overall improvement in public health and the reduction of medical costs associated with these diseases, more than compensate for the initial infrastructure investment. Among the set of policies analyzed, improving access to piped water is the most promising, given its immediate impact and its potential to generate positive secondary effects on public health. However, to definitively support this conclusion, it would be necessary to conduct a more detailed cost-benefit analysis, which would allow for a precise quantification of the net benefits derived from this intervention. The explanation provided here has been incorporated into the updated article on page 15, second paragraph, line 515. |

Reviewer 2 Report
Comments and Suggestions for Authors
line 291 where
Check other such instances
You have a nice interesting paper. There are of course studies showing that access to health care depends on income in high and low-income countries, So the problem is more general. Perhaps you want to add observations on access to schooling in Mexico. Why are parents not better informed? Or this is a matter of living conditions and culture. It would be helpful to know more.
Comments on the Quality of English LanguageEnglish expression is very good.
Author Response
|
Comments 1: [You have a nice interesting paper. There are of course studies showing that access to health care depends on income in high and low-income countries, So the problem is more general. Perhaps you want to add observations on access to schooling in Mexico. Why are parents not better informed? Or this is a matter of living conditions and culture. It would be helpful to know more.]
|
|
Response 1: [Thank you for pointing this out. We greatly appreciate your valuable observation regarding the relationship between income, access to healthcare, and education. Our study has identified that marginalization conditions in Mexico, particularly in the states of Chiapas, Oaxaca, and Guerrero, are key factors that not only affect access to healthcare and education but also to essential resources like potable water. These structural issues go beyond income and require a comprehensive approach. Regarding your comment on parental education, we fully agree that it is a critical area to improve child health. Our analysis highlights that improving parental education and the infrastructure for basic services, such as access to potable water, are essential interventions to reduce child mortality in these marginalized areas. Additionally, the cultural and linguistic barriers faced by indigenous communities further limit access to healthcare, emphasizing the need to develop culturally inclusive health services. Although this paper does not directly address access to healthcare or education in schools, we believe these lines of analysis represent an excellent opportunity for future research.]
|

Reviewer 3 Report
Comments and Suggestions for Authors
This is an insightful and important look at the impact of socioeconomic and cultural factors on child mortality outcomes. By evaluating the drivers for waterborne gastrointestinal diseases and resulting health disparities among children of Mexico, the authors were to determine key areas for policy recommendations and interventions to prevent disease. The information provided in this analysis will likely generate great discussion among public health professional and national policy makers, hopefully sparking real change to protect the health of children and their families in vulnerable settings. The reviewer does note several areas in which edits could help the reader grasp the drivers/variables and to better understand the parameters used in their determination.
Introduction
· Line 56: The word ‘Several’ is capitalized
· Please add any time restrictions used in the literature review to the description here and also in the Table 1 title (ex. Publications Between XXXX and XXXX). This will allow researchers to know where to begin follow up studies in the future.
· Table 1: Would be helpful to use a bold font or some other way to illustrate the country names in the first column of the table. The country names get lost in the same line as the citation
· Table 1: The study type is missing for Galiani et al. [8] under the Latin America section
· Table 1: The column for Study Type list the statistical methods used for some studies instead of the study type. For example, Mebrahtom et al. [11] is listed as a descriptive study. But this is likely a cross-sectional descriptive study, like Zuta Arriola et al. [5] Recommend renaming this column to represent both study types and statistical methods if the study type could not be determine for all included publications. Alternatively, add another column for statistical method used.
· Table 1: For the region of the United States, this is the country name but not the region name. Recommend changing the region name to North America and then replacing Estados Unidos with United States.
· Table 1: Recommend moving the Global section to the end, after the North America region
· Table 1: Please include a column with the main finding(s) from each study.
· Line 63: Here it says that the study data is presented by methodological approach, but this differs from the column title Study Type. See recommendation above.
· Line 67: The wording, “various variables” could be rewritten for clarity
· Lines 72-73: Here the authors are talking about the findings from several studies, but this is not presented to the reader in Table 1 without an additional column of the main findings from each study. Unless this is what was meant by Key Variables? But this reviewer took that to mean the variables that were examined by the study.
· Across the section of the introduction that is describing the studies, it isn’t clear whether the factors being presented are regarding communities, households, or individuals. For example, Line 74, are the socioeconomic and educational factors at the household level? Community level? Or maternal v. paternal levels? Again, Line 80, there is a lack of knowledge presented as associated with WGD in Latin America, but the reviewer isn’t sure who has this lack of knowledge. Recommend revising the review of the literatures’ main findings in the introduction to add more details on who/where the factor is located across community, household, and individual domains.
· Lines 83-85: Is there a reason provided as to why the privatization reduced mortality? Did the privatization improve the water quality and safety? Or allow more reliable sources? An explanation for why this positive impact occurred would be helpful here.
· Line 86: Water storage in the household?
· Line 92: Please provide details for the type of environmental factors
· Line 108: Which approaches have been essential? Many were outlined in the table.
· Line 127: Do the authors mean region here? Or the younger population?
· Line 140: Which high-risk communities? Is this the 3-15 year olds?
Materials and Methods
· Figure 1 caption: Include the year for the child mortality data
· Line 185: Add the year here, too, for when the 154 deaths occurred
· Lines 204-205: What indicators are used to determine marginalization using this index? Please provide more background for the reader on how households and locations are classified as marginalized.
Results
· Line 281: Recommend revising the phrase “curable” diseases to “preventable” diseases
· Line 281: Recommend revising ‘causes’ to something like, ‘identify the associations between WGD and child mortality….’
· Is y all gastrointestinal diseases? Or WGD? Line 304 lists the dependent variable as WGD but line 292 and Table 3 use the term gastrointestinal diseases. Be consistent.
· Table 3: What is the authors’ definition of an “overcrowded household”? Please provide these details.
· Table 3: What is the authors’ definition of “basic education”? Please provide these details.
· Line 419 and throughout: Is this piped water to the household or premises? Or either?
· Table 6: Heading should probably be plural (ex. Strategies); Remove extra period/dot
· Table 6: Fix spacing/formatting issues in column 1
· Lines 449-487 and possibly Table 6 in line for total impact: The reviewer recommends adding the estimated lives saved by each of the anticipated reduction percentages for each policy to strengthen the authors’ message.
Discussion
· Lines 582-584: The reviewer wonders what other relationships exist when examining culture and health outcomes. For example, are indigenous language speakers also in households that have disproportionately lower rates of piped water? Or overcrowding etc.? What do the marginalization rates within the marginalized locations look like?
Conclusion
· Shorten to one paragraph or just over a paragraph. Move the other text presented here to the Discussion section above.
Author Response
|
Comments 1: [Introduction · Line 56: The word ‘Several’ is capitalized · Please add any time restrictions used in the literature review to the description here and also in the Table 1 title (ex. Publications Between XXXX and XXXX). This will allow researchers to know where to begin follow up studies in the future. · Table 1: Would be helpful to use a bold font or some other way to illustrate the country names in the first column of the table. The country names get lost in the same line as the citation · Table 1: The study type is missing for Galiani et al. [8] under the Latin America section · Table 1: The column for Study Type list the statistical methods used for some studies instead of the study type. For example, Mebrahtom et al. [11] is listed as a descriptive study. But this is likely a cross-sectional descriptive study, like Zuta Arriola et al. [5] Recommend renaming this column to represent both study types and statistical methods if the study type could not be determine for all included publications. Alternatively, add another column for statistical method used. · Table 1: For the region of the United States, this is the country name but not the region name. Recommend changing the region name to North America and then replacing Estados Unidos with United States. · Table 1: Recommend moving the Global section to the end, after the North America region · Table 1: Please include a column with the main finding(s) from each study. · Line 63: Here it says that the study data is presented by methodological approach, but this differs from the column title Study Type. See recommendation above. · Line 67: The wording, “various variables” could be rewritten for clarity · Lines 72-73: Here the authors are talking about the findings from several studies, but this is not presented to the reader in Table 1 without an additional column of the main findings from each study. Unless this is what was meant by Key Variables? But this reviewer took that to mean the variables that were examined by the study. · Across the section of the introduction that is describing the studies, it isn’t clear whether the factors being presented are regarding communities, households, or individuals. For example, Line 74, are the socioeconomic and educational factors at the household level? Community level? Or maternal v. paternal levels? Again, Line 80, there is a lack of knowledge presented as associated with WGD in Latin America, but the reviewer isn’t sure who has this lack of knowledge. Recommend revising the review of the literatures’ main findings in the introduction to add more details on who/where the factor is located across community, household, and individual domains. · Lines 83-85: Is there a reason provided as to why the privatization reduced mortality? Did the privatization improve the water quality and safety? Or allow more reliable sources? An explanation for why this positive impact occurred would be helpful here. · Line 86: Water storage in the household? · Line 92: Please provide details for the type of environmental factors · Line 108: Which approaches have been essential? Many were outlined in the table. · Line 127: Do the authors mean region here? Or the younger population? · Line 140: Which high-risk communities? Is this the 3-15 year olds?]
|
||||
|
Response 1: [Thank you for pointing this out. We greatly appreciate the detailed feedback provided. Below, we address each of the comments and explain the corresponding changes made to the manuscript. Line 56: The capitalization error in "Several" has been corrected. Time Restrictions in Literature Review: We have added the time restrictions used in the literature review to both the introduction and the title of Table 1. This ensures that future researchers have a clear starting point for follow-up studies. Table 1 – Country Names: In Table 1, we have now bolded the country names to distinguish them more clearly from the citations. This enhances readability and makes it easier to locate the countries in the first column. Study Type for Galiani et al. [8]: We have added the study type for Galiani et al. [8], noting that it is a quasi-experimental study under the Latin America section. Study Type and Statistical Methods: Following your suggestion, we renamed the column to Methodological Approaches (MA) and Statistical Methods (SM) to better represent both the study types and statistical methods used. This change ensures that readers can more easily distinguish between the methodological approach and the statistical techniques applied. Region Name: We have updated the region name from Estados Unidos to North America, and changed "Estados Unidos" to United States to align with your recommendation. Reorganization of the Global Section: As suggested, we have moved the Global section to the end of the table, following the North America region for better flow and clarity. Main Findings Column: We have added a new column titled Main Findings in Table 1, providing a brief summary of the key results from each study. This offers a more comprehensive view of the studies for the readers. Line 63: We have updated the text to reflect the new column title, Methodological Approaches (MA) and Statistical Methods (SM), for consistency between the table and the text. Line 67: The phrase "various variables" has been rewritten by “examine a range of factors”, now specifying the exact variables examined in the studies. Line 72-73 – Main Findings: We have clarified the difference between Key Variables and Main Findings in both the introduction and Table 1. The new Main Findings column now presents the key outcomes of each study, while Key Variables refers to the variables examined in the studies. Lines 74-80 Community, Household, or Individual Levels (Lines 74-80): In response to your feedback, we have opted to explicitly indicate the unit of statistical observation in the Key Variables column for each study in Table 1, making it clear whether the analysis focuses on communities, households, or individuals. Additionally, lines 74 and 80 in the introduction have been revised to clarify who the observations refer to, ensuring there is no ambiguity. Lines 83-85 – Privatization and Mortality: In line 87 We have added an explanation as to why water privatization in the Galiani et al. [8] study reduced mortality, noting that privatization improved water quality, safety, and reliability of water supply. Line 86 – Water Storage: In line 92 We have clarified that this refers to water storage at the household level, addressing the ambiguity. Line 92 – Environmental Factors: In line 94 tThe term "environmental factors" has been clarified by “quality of water sources”. Line 108 – Essential Approaches: In lines 103 to 111 We have specified which approaches outlined in the table were deemed essential, clarifying their importance to the findings. Line 127 – We have clarified that the reference is about younger population. Line 140 – We have clarified that the high-risk refers children aged 0-3 years.
|
||||
|
Comments 2: [Materials and Methods · Figure 1 caption: Include the year for the child mortality data · Line 185: Add the year here, too, for when the 154 deaths occurred · Lines 204-205: What indicators are used to determine marginalization using this index? Please provide more background for the reader on how households and locations are classified as marginalized.] |
||||
|
Response 2: Figure 1 caption: We have updated the caption to include the year for the child mortality data, specifying that the data corresponds to 2019.
Line 185: In line 191 We have added the year 2019 to clarify when the 154 deaths occurred, ensuring consistency throughout the manuscript.
Lines 204-205: In line 212 We have provided additional background on the indicators used to determine marginalization according to the Marginalization Index (Índice de Marginación) by CONAPO. The index is based on several key indicators, including education level, housing conditions, income levels, and access to basic services. We have expanded the explanation to include how households and locations are classified as marginalized, using these variables to assess levels of social and economic deprivation. This information has been incorporated to provide readers with a more comprehensive understanding of how marginalization is measured.
|

Round 2
Reviewer 1 Report
Comments and Suggestions for Authors
No further comments.